## RESEARCH ARTICLE

# Multiomics analysis of GSTP1 knockdown pancreatic cancer cells reveals key regulators of redox and metabolic homeostasis

Jenna N. Duttenhefner*, Rahul R. Singh, Katherine Schmidt and Katie M. Reindl

## ABSTRACT

Glutathione S transferase pi-1 (GSTP1) is a detoxification enzyme essential for oxidative homeostasis. In cancer, GSTP1 has been implicated in tumorigenicity, cell cycle progression, and chemoresistance. While GSTP1 depletion has been associated with decreased cancer growth in various models, the mechanism remains poorly understood. This study investigates GSTP1 as a therapeutic target for pancreatic ductal adenocarcinoma (PDAC) using inducible knockdown models. We demonstrate that GSTP1 loss disrupts redox balance, impairs cell survival, and induces metabolic adaptations. Multiomics analysis characterized the global impact of inducible GSTP1 knockdown on the transcriptome and proteome of PDAC cells, identifying 550 differentially expressed genes and 62 proteins. Notably, 43 of these showed consistent regulation at both the mRNA and protein levels. We identify dysregulation of key stress response proteins, including dimethylarginine dimethylaminohydrolase 1 (DDAH1), involved in nitric oxide metabolism, and protein disulfide isomerase A6 (PDIA6), which maintains protein homeostasis. The interplay between GSTP1, DDAH1 and PDIA6 highlights the complexity of redox regulation in pancreatic cancer and suggests that targeting GSTP1 may offer a new therapeutic approach for PDAC.

KEY WORDS: Pancreatic ductal adenocarcinoma, GSTP1, Multiomics analysis, Redox homeostasis, Therapeutic targets

## INTRODUCTION

Pancreatic ductal adenocarcinoma (PDAC) poses a significant challenge in oncology due to its aggressive nature and limited treatment options. With over 67,000 projected cases of pancreatic cancer in the USA in 2025, this malignancy persists as one of the most prevalent cancers, accompanied by a dishearteningly low 5-year survival rate of only 13% (Siegel et al., 2025). Despite extensive research efforts, the incidence and mortality rates have continued to increase (Siegel et al., 2025). These rates, along with compounding factors such as limitations for early diagnosis, rapid metastasis resulting in restrictive access to curative surgery, and resistance to conventional chemotherapy approaches, drive the projection that pancreatic cancer will become the second-leading cause of

Department of Biological Sciences, North Dakota State University, Fargo, ND 58102, USA.

*Author for correspondence ( jenna.duttenhefner@ndsu.edu)

(ID) J.N.D., 0009-0007-2288-5582; R.R.S., 0000-0001-5572-8216; K.M.R., 0000-0002-7880-3845

cancer-related mortality by 2030 (Barcellini et al., 2020; Park et al., 2021). The current standard of care for patients with locally advanced and metastatic PDAC includes continuous chemotherapy with either FOLFIRINOX (leucovorin, 5-fluorouracil, irinotecan, and oxaliplatin) or a gemcitabine/nanoparticle albumin-bound-paclitaxel combination treatment (Sohal et al., 2016). Unfortunately, these treatments have only marginal effects on overall clinical outcomes, underscoring the need for advances in understanding the unique adaptations of pancreatic cancer cells compared to normal cells, in order to develop novel therapeutic strategies that target this aggressive disease (Parrasia et al., 2020).

PDAC thrives in a hostile tumor microenvironment characterized by hypoxia and nutrient deprivation (Kamphorst et al., 2015; Koong et al., 2000). Cancer cells undergo metabolic rewiring to adapt to these conditions, including molecular and physiological alterations in their antioxidant defense mechanisms, which help maintain redox homeostasis (Guillaumond et al., 2015, 2013; Ying et al., 2012). Redox homeostasis is a precise and complex balance between reactive oxygen species (ROS) and antioxidants that all cells perform to maintain normal cellular function and survival. Compared with normal cells, cancer cells have increased metabolic activity to sustain their rapid growth and proliferation, leading to the accumulation of ROS. High ROS levels can induce oxidative stress, leading to severe cellular damage and the activation of cell death pathways (Diehn et al., 2009). Cancer cells must adapt to this stress phenotype for survival, leading to a dependency on enhanced antioxidant activation to scavenge excess ROS (Gorrini et al., 2013; Xing et al., 2022). The reliance of cancer cells on these altered antioxidant systems presents an opportunity for selective therapeutic targeting of cancer cells. The induction of oxidative stress and impairment of antioxidant response function as mechanisms of action for various drug and cancer therapies (Gorrini et al., 2013).

Glutathione S-transferase pi-1 (GSTP1) is a key player in the antioxidant system due to its role in conjugating reactive electrophiles to glutathione (Bakhiya et al., 2007). GSTP1 is overexpressed in multiple cancers (Howells et al., 2004; Ruzza et al., 2009; Wang et al., 2022; Ye et al., 2006), including PDAC, where its high expression is associated with tumor progression and poor clinical outcomes (Singh et al., 2020). Beyond its canonical role in detoxification, GSTP1 has also been implicated in cancer cell metabolism (Louie et al., 2016), apoptotic signaling (Adler, 1999; Wu et al., 2006), and drug resistance (Singh and Reindl, 2021). Our previous research on a constitutive GSTP1 knockdown system demonstrated that disrupting the antioxidant system by targeting GSTP1 in PDAC cells leads to increased ROS levels and apoptotic cell death (Singh et al., 2020). However, the precise mechanisms underlying GSTP1's role in promoting pancreatic cancer cell growth and proliferation remain unclear.

To address this gap, we developed novel doxycycline-inducible GSTP1-knockdown PDAC cell models. Using multiomics techniques, we investigated the global impact of inducible GSTP1

knowledge on the transcriptome and proteome of PDAC cells. The transcriptome and proteome reflect a specific metabolic state of PDAC cells as they adapt and shift their metabolic activity and cellular signaling. Insights into these dynamic systems can provide valuable knowledge on the molecular changes and mechanisms underlying PDAC progression. Our findings reveal significant alterations in the transcriptomic and proteomic profiles of PDAC cells following inducible knockdown of GSTP1. Measuring these changes will help elucidate the interplay between GSTP1 and pathological processes in PDAC progression. Moreover, we observed that redox status is a likely driver of transcriptional regulation in our GSTP1 knockdown cells. Given GSTP1's pivotal role in maintaining redox homeostasis, targeting GSTP1 and its associated metabolic pathways presents promising therapeutic opportunities for PDAC treatment.

## RESULTS
### Establishment of doxycycline-inducible GSTP1-knockdown PDAC cells
To investigate the role of GSTP1 in PDAC, we generated doxycycline-inducible GSTP1 knockdown cell lines (MIA PaCa-2, PANC-1, and HPAF-II) using SMARTvector™ lentiviral constructs containing either non-specific (NS) control or GSTP1-specific shRNA (shGSTP1-1 and shGSTP1-2) sequences (Fig. 1A). The NS control shRNA showed no significant homology to any known human, mouse, or rat genes. Western blotting and qRT-PCR analyses confirmed that doxycycline treatment led to a >90% reduction in GSTP1 expression at both the protein and mRNA levels after 96 h (Fig. 1B-D). Control siRNA did not affect GSTP1 expression. Expression levels were restored upon doxycycline withdrawal, demonstrating the reversible nature of the knockdown system (Fig. 1E-M). This model enabled a controlled investigation into the effects of GSTP1 depletion on the function of PDAC cells.

### GSTP1 knockdown reduces PDAC cell growth and increases ROS levels
MTT assays revealed a significant reduction in proliferation across all three PDAC cell lines following knockdown, with MIA PaCa-2 and PANC-1 exhibiting >20% reductions and HPAF-II >15% (Fig. 2A). In previous studies, constitutive GSTP1 knockdown was shown to reduce cell viability and proliferation in pancreatic (Singh et al., 2020) and other cancers (Checa-Rojas et al., 2018; Hokaiwado et al., 2008; Louie et al., 2016; Mutallip et al., 2011; Tusskorn et al., 2018). GSTP1 plays a crucial role as an antioxidant enzyme, maintaining the optimal redox environment within cells. Given GSTP1's role in oxidative homeostasis, we measured intracellular ROS levels using CellROX™ staining. Flow cytometry analysis showed a 2.1-fold, 1.5-fold, and 1.7-fold increase in ROS levels in MIA PaCa-2, PANC-1, and HPAF-II cells, respectively, following knockdown. ROS levels returned to near-control levels upon 48 h of doxycycline withdrawal, confirming the reversible impact of GSTP1 expression on oxidative balance (Fig. 2B-C).

### GSTP1 knockdown induces a differential transcriptomic response in PDAC cells
RNA sequencing (RNA-seq) was performed to understand the influence of reduced GSTP1 expression on the PDAC transcriptome and to identify the potential underlying mechanisms of impaired growth in response to increased oxidative stress. A total of 211,535,272 single-end, 150 base pair reads were obtained from eight samples (four biological replicates each of NS control and GSTP1 knockdown MIA PaCa-2 cells). An average of 26,441,909

high-quality reads were obtained for each sample. More than 96% of the reads mapped successfully to the human genome, and approximately 90% were aligned exclusively to unique regions. The characteristics of the output reads are summarized in Table S1. A high correlation was observed between all four biological replicates of each sample, as represented by principal component analysis (PCA) (Fig. 3A). A total of 2344 genes showed significant (P-adj <0.05) changes in expression between the NS control and the GSTP1 knockdown MIA PaCa-2 cells. Of these, RNA sequencing identified 550 differentially expressed genes, with 237 upregulated and 313 downregulated in GSTP1 knockdown MIA PaCa-2 cells compared to NS controls (Fig. 3B). The top differentially expressed genes were visualized using the enhanced heatmap tool in the gplots library (Fig. 3C-E).

### GSTP1 knockdown induces a differential proteomic response in PDAC cells
To determine whether the transcriptomic results were reflected at the protein level, we next investigated the effects of inducible GSTP1 knockdown on the global proteomic signature of PDAC cells. In total, 5871 unique proteins were identified in the MIA PaCa-2 NS control and GSTP1 knockdown cells. A high correlation was observed between all five biological replicates of each sample, as represented by principal component analysis (PCA) (Fig. 4A). Proteomics analysis identified 62 differentially expressed proteins, with 16 upregulated and 46 downregulated in GSTP1 knockdown cells compared to control cells (Fig. 4B). These differentially expressed proteins were visualized using the enhanced heatmap tool in the gplots library (Fig. 4C-E).

### RNA sequencing and proteomics experiments reveal parallel responses to GSTP1 knockdown
A comparative analysis of our bulk RNA sequencing and LC-MS/MS-based proteomics experiments suggests a coordinated regulatory network following GSTP1 knockdown. Among the 550 genes and 62 proteins significantly and differentially expressed in GSTP1 knockdown MIA PaCa-2 cells compared to the control, we found 43 genes that were upregulated or downregulated at the mRNA or protein level (Fig. 5A). To assess the changes in the cellular and molecular pathways associated with GSTP1 knockdown in PDAC cells, we performed functional pathway analysis using Ingenuity Pathway Analysis (Qiagen) software. Pathway analysis identified significant dysregulation in cellular function, maintenance, and signaling pathways following GSTP1 knockdown. Among these pathways, notable alterations were observed in metabolic regulation, extracellular matrix interactions, focal adhesion, and nitric oxide metabolism, reinforcing GSTP1's broad influence on cellular homeostasis (Fig. 5B). Similarly, gene set enrichment analysis using Enrichr revealed that genes involved in metabolism, oxidative phosphorylation, and cellular signaling were differentially expressed in GSTP1 knockdown cells compared with the control. The significantly enriched pathways identified in our comparative transcriptomics and proteomics experiments are summarized in Tables 1 and 2, respectively.

Dimethylarginine dimethylaminohydrolase 1 (DDAH1) and protein disulfide isomerase A6 (PDIA6) exhibited consistent regulation at both the mRNA and protein levels. DDAH1, involved in nitric oxide (NO) metabolism, was upregulated, while PDIA6, critical for protein folding and redox balance, was downregulated. Validation using qPCR and western blotting in control and GSTP1 knockdown PDAC cells confirmed these changes. DDAH1 was significantly upregulated in GSTP1 knockdown cells across all three

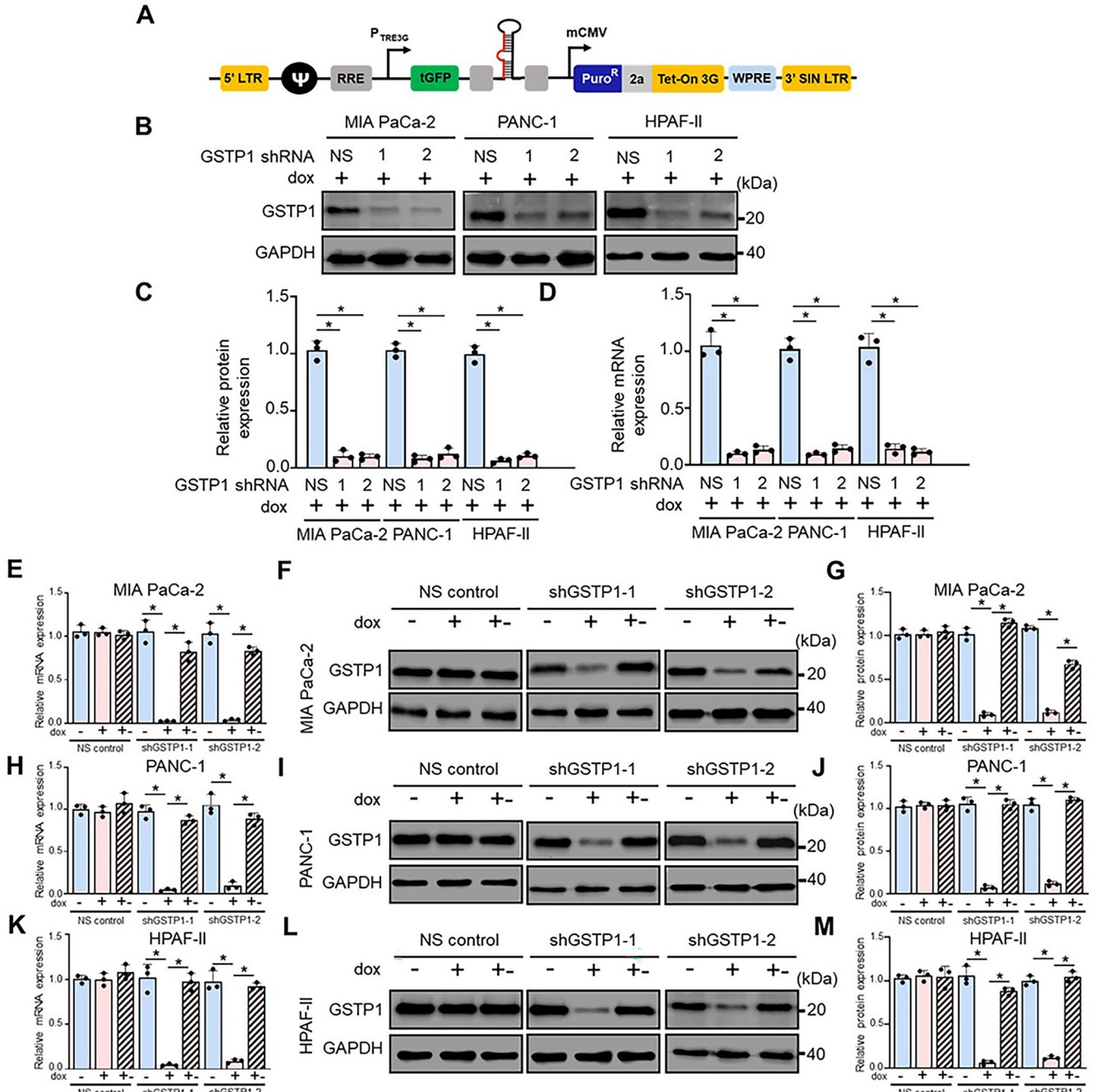

**Fig. 1. Establishing doxycycline-inducible GSTP1 knockdown in PDAC cells.** (A) Schematic of the Tet-inducible SMART™ vector used to express GSTP1 shRNAs (shGSTP1-1 and shGSTP1-2) in MIA PaCa-2, PANC-1, and HPAF-II cells. GSTP1 knockdown was confirmed by (B,C) western blotting and (D) qRT-PCR after 96 h of 500 ng/ml doxycycline (dox) treatment. To restore GSTP1 expression, dox was removed for 120 h (dox +-), and qPCR and protein levels were re-evaluated by western blotting in MIA PaCa-2 (E-G), PANC-1 (H-J), and HPAF-II (K-M) cells. β-actin and β-tubulin were used as housekeeping genes for normalization of all qRT-PCR data. All western blot data were normalized to GAPDH, and protein/mRNA levels from NS control cells were compared with those from shGSTP1-1 and shGSTP1-2. Images represent three independent experiments (*n*=3). A Student's *t*-test was used to assess significance, with * denoting *P*<0.05. Error bars represent standard deviation.

PDAC lines, with MIA PaCa-2 showing over 100% increase in mRNA and nearly 90% at the protein level (Fig. 5C-E). PANC-1 and HPAF-II cells exhibited a similar trend, with substantial increases in both mRNA and protein expression, though the extent varied across cell lines (Fig. S1A-F). Conversely, PDIA6 was consistently downregulated, with MIA PaCa-2 showing over 70% decreases at both mRNA and protein levels (Fig. 5C-E). In PANC-1 cells, mRNA levels dropped by more than 60%, with protein levels decreasing by

over 80% (Fig. S1A-C). HPAF-II exhibited a more moderate decline, with mRNA reduced by about 45% and protein by over 50% (Fig. S1D-F). This opposing regulation suggests a complex interplay between oxidative stress adaptation and protein homeostasis. The differential expression of DDAH1 and PDIA6 in GSTP1 knockdown PDAC cells is particularly intriguing, as both are involved in redox homeostasis and are frequently upregulated in various cancers (Wang et al., 2023; Ye et al., 2017). While experimental evidence for specific

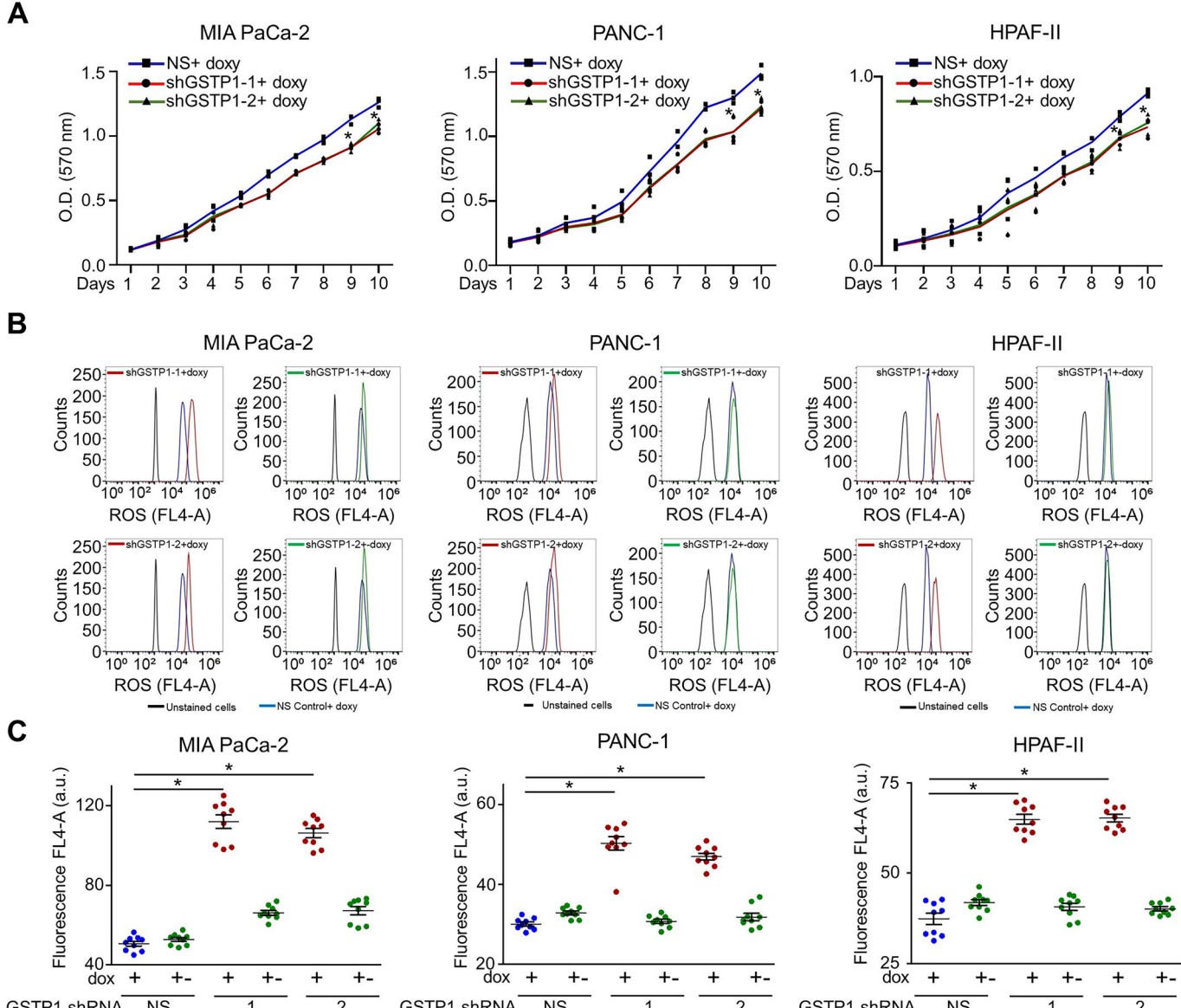

**Fig. 2. GSTP1 knockdown reduces PDAC cell growth and increases ROS levels.** (A) MTT assays demonstrate a significant decrease in cell proliferation following GSTP1 knockdown in MIA PaCa-2, PANC-1, and HPAF-II cells over a 10-day period, with absorbance (570 nm) measured every 24 h. The results are averages from three experiments with 24 technical replicates. (B) Flow cytometry analysis of ROS using CellROX™ DeepRed staining, comparing NS control (NS) and GSTP1 knockdown cells (shGSTP1-1 and shGSTP1-2) and GSTP1 recovery after 120 h of doxycycline (dox) removal (+-). (C) Quantification of ROS levels shows comparisons between NS control and two independent shGSTP1 sequences (shGSTP1-1, shGSTP1-2). Data are representative of three experiments (*n*=3), and statistical significance was evaluated using a Student's *t*-test (*$P$<0.05). Error bars represent standard deviation.

signaling pathways remains to be established, the observed changes likely reflect broader redox-regulated cellular responses.

To further expand our understanding of the mechanism through which GSTP1 knockdown influences the expression of these genes, we restored GSTP1 expression by removing doxycycline from the culture media and re-evaluated the mRNA and protein expressions of PDIA6 and DDAH1. The baseline significance between control and knockdown conditions was previously established (Fig. 5C-E; Fig. S1A-F) and is not repeated here to maintain clarity. Upon restoring GSTP1 expression, the mRNA and protein levels of both PDIA6 and DDAH1 returned to near-control levels in MIA PaCa-2 cells, with DDAH1 dropping below baseline at the protein level (Fig. 5F-H). Similar results were observed in the PANC-1 and HPAF-II cell lines (Fig. S2A-F), though mRNA and protein

DDAH1 levels were significantly reduced below baseline in both PANC-1 and HPAF-II cells.

## Evidence of redox-dependent expression regulation in GSTP1 knockdown cells

To determine whether oxidative stress mediates these expression changes, we treated GSTP1 knockdown cells with antioxidants, including N-acetylcysteine (NAC), glutathione (GSH), and Vitamin E. NAC is a direct scavenger of ROS and a reducing agent capable of cleaving disulfide bonds. The principal role of NAC as an antioxidant lies in its role as a precursor of L-cysteine, which is the rate-limiting step in glutathione synthesis (Aldini et al., 2018; Dodd et al., 2008). Glutathione is the most abundant endogenous antioxidant and can directly scavenge a diverse range of ROS to prevent oxidative damage.

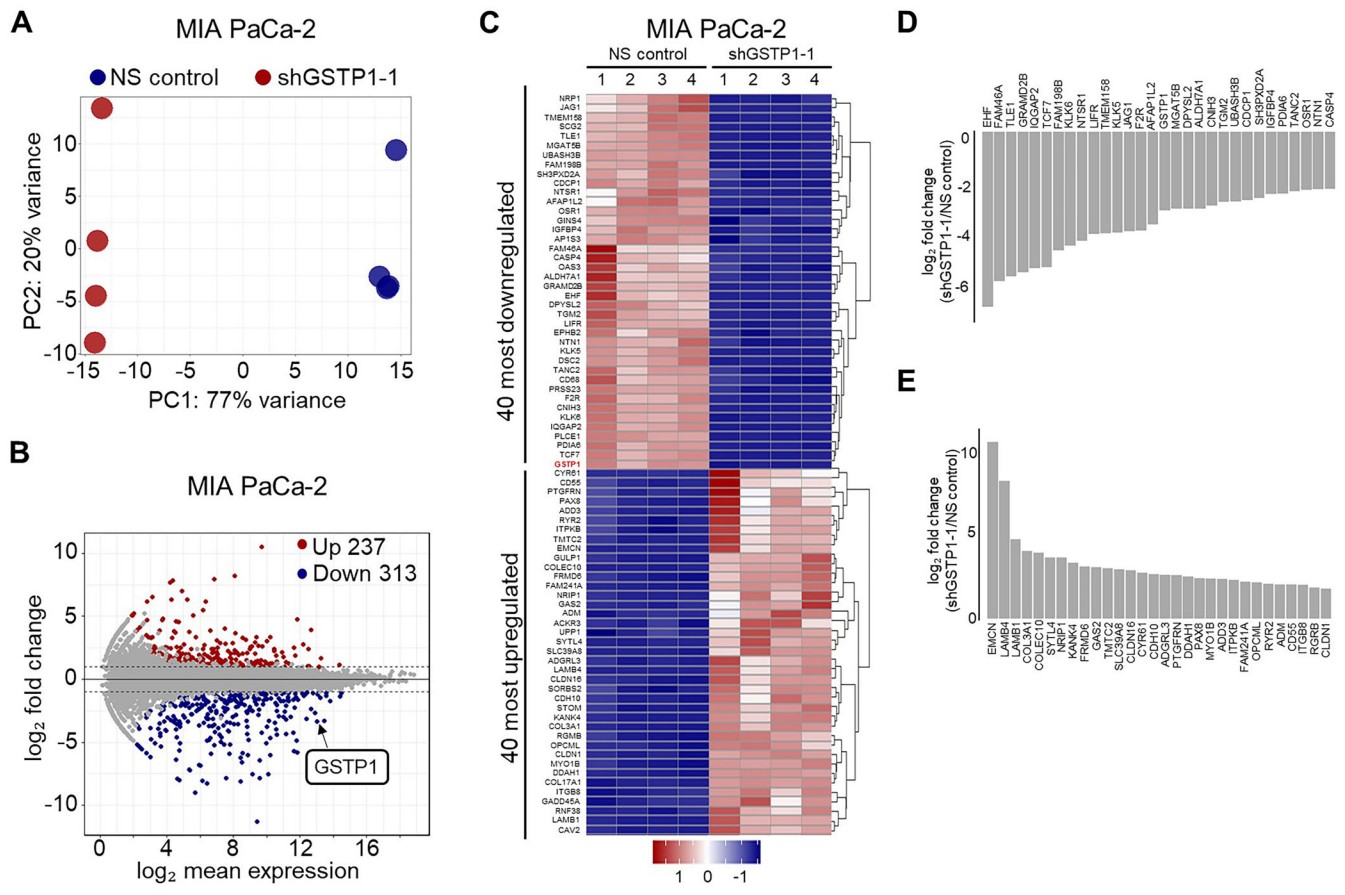

**Fig. 3. GSTP1 knockdown induces a differential transcriptome in PDAC cells.** (A) Principal component analysis (PCA) plot illustrating distinct clustering between NS control and GSTP1 knockdown (shGSTP1-1) transcriptome in MIA PaCa-2 cells ($n$=4). (B) MA-plot showing upregulated (red) and downregulated (blue) genes (padj<0.01 and log$_2$-fold<−1 or>1) in GSTP1 knockdown cells compared to the NS control. (C) Heatmap of the 40 most significantly dysregulated genes. The dark blue color indicates the downregulation of genes in GSTP1 knockdown MIA PaCa-2 cells compared to the control, while red indicates the upregulation of genes. (D,E) Bar graphs displaying the 30 most significantly (D) downregulated and (E) upregulated genes. Data are from RNA-Seq analysis with four biological replicates.

Glutathione is also involved in detoxifying reactive intermediates through the activity of glutathione S-transferases and as a cofactor of glutathione peroxidases (Forman et al., 2009). Glutathione can also indirectly regenerate Vitamin E through a Vitamin C-dependent antioxidant network of enzymatic and non-enzymatic mechanisms (Packer et al., 2001). Vitamin E is a lipid-soluble, radical-scavenging antioxidant that primarily protects cell membranes and lipoproteins from oxidative damage (Packer et al., 2001). Fig. 6A provides an overview of key direct and indirect mechanisms underlying the antioxidant properties of NAC, glutathione, and Vitamin E.

DDAH1 expression was significantly reduced at both the mRNA and protein levels following antioxidant treatment in MIA PaCa-2 GSTP1 knockdown cells, suggesting its upregulation as a compensatory response to oxidative stress (Fig. 6B-J). Similarly, in PANC-1 cells, treatment with NAC, GSH, and Vitamin E significantly reduced the levels of DDAH1 at both the mRNA and protein levels (Fig. S3A-I). In HPAF-II GSTP1 knockdown cells, NAC treatment significantly reduced DDAH1 mRNA, while all treatments led to significant reductions in DDAH1 protein expression (Fig. S4A-I).

Interestingly, in MIA PaCa-2 GSTP1 knockdown cells, treatment with 5 mM NAC or 5 mM Vitamin E for 48 h significantly increased PDIA6 mRNA levels, although this effect did not extend to protein expression (Fig. 6B-D, H-J). In contrast, GSH treatment did not

significantly affect PDIA6 mRNA or protein levels in MIA PaCa-2 cells (Fig. 6E-G). In PANC-1 cells, only NAC treatment resulted in a significant increase in PDIA6 mRNA expression, with no change in protein expression observed with any antioxidant treatment (Fig. S3A-I). In HPAF-II GSTP1 knockdown cells, PDIA6 expression was not significantly altered by any antioxidant treatment at either the mRNA or protein level. The observed discrepancy in the regulation of these proteins across cell lines may reflect differences in baseline oxidative stress levels and redox adaptation mechanisms.

To determine whether antioxidant treatment could rescue the reduced viability phenotype associated with GSTP1 knockdown, MIA PaCa-2 cells were treated with NAC, Vitamin E, or GSH during the final 48 h of a 96-h doxycycline induction period. Cell proliferation, as measured by MTT at the 96-h time point, was largely unchanged across most treatment conditions compared to control cells without doxycycline treatment (Fig. S5A). However, two conditions showed small but statistically significant deviations. NS control cells treated with GSH or Vitamin E exhibited slightly reduced proliferation. In contrast, GSTP1 knockdown cells without doxycycline showed a modest increase. The reduced proliferation observed in antioxidant-treated control cells may reflect an overcorrection of the redox environment, where dampening ROS below physiological levels impairs redox-sensitive proliferative signaling. Conversely, the increase in proliferation observed in the knockdown cells without

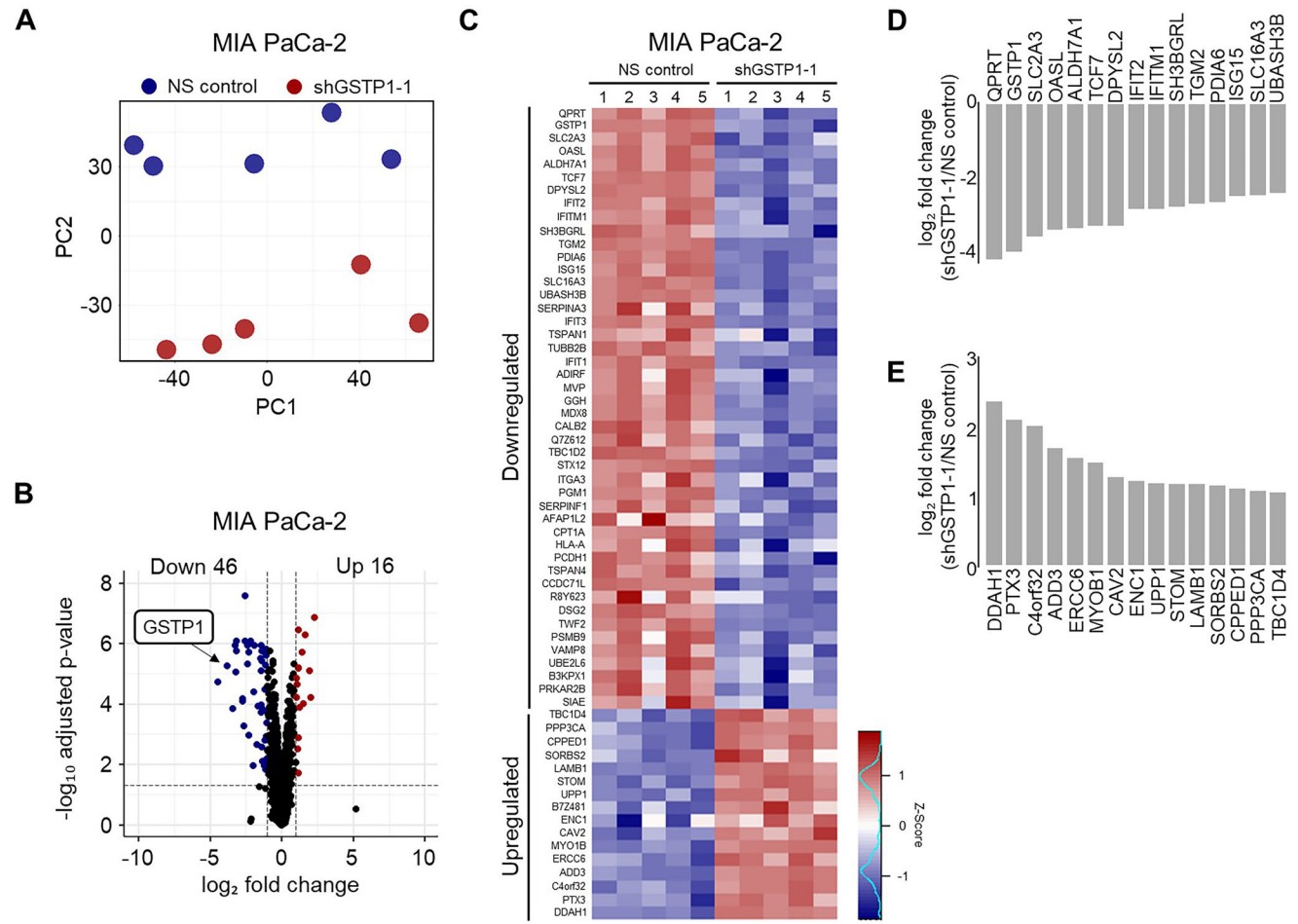

**Fig. 4. GSTP1 knockdown alters the global proteomic signature in PDAC cells.** (A) PCA plot illustrating distinct clustering between GSTP1 knockdown (shGSTP1-1) compared to NS control MIA PaCa-2 cell proteomic signatures (*n*=5). (B) Volcano plot of upregulated (red) and downregulated (blue) proteins (padj<0.01 and log$_2$-fold<−1 or>1) in GSTP1 knockdown (shGSTP1-1) cells compared to the NS control. (C) Heatmap of the 52 most downregulated and 17 most upregulated proteins. The dark blue color indicates downregulated proteins in GSTP1 knockdown MIA PaCa-2 cells compared to the control, and red indicates upregulated proteins. (D-E) Bar graphs of the 15 most significantly (D) downregulated and (E) upregulated proteins in GSTP1 knockdown cells. Data are from LC-MS/MS proteomics with five biological replicates.

doxycycline may suggest the presence of compensatory mechanisms in the absence of GSTP1 suppression, although the biological relevance of this effect remains unclear. Overall, these findings suggest that redox balance, rather than ROS suppression alone, plays a nuanced role in regulating PDAC cell proliferation.

To validate that antioxidant treatment was effective in mitigating ROS, MIA PaCa-2 NS control and shGSTP1-1cells were treated with doxycycline for 96 h with or without co-treatment with 5 mM NAC during the final 48 h. ROS levels were assessed using CellROX$^{TM}$ DeepRed fluorescence and flow cytometry. As shown in the representative histogram (Fig. S5B), NAC treatment substantially recovered ROS accumulation in GSTP1 knockdown cells. Quantification of FL4-A fluorescence intensities confirmed that the ROS levels in GSTP1 knockdown cells treated with NAC were restored to levels comparable to those of untreated control cells (Fig. S5C). These findings validate the functional activity of NAC in this model and confirm that antioxidant treatment successfully reversed the redox phenotype associated with GSTP1 suppression.

## DISCUSSION

Pancreatic cancer is a challenging disease with limited treatment options and a poor prognosis. GSTP1, a phase II metabolic enzyme, plays a significant role in detoxifying carcinogens and cytotoxic drugs through glutathione conjugation (Arai et al., 2006). Beyond its classical detoxification function, GSTP1 has been implicated in tumorigenicity, oxidative stress protection, and cell cycle regulation in pancreatic cancer (Dang et al., 2005). Moreover, it has been identified as an autocrine stimulator of mutated KRAS signaling, reinforcing its potential as a target for chemoprevention in pancreatic cancer (Kogawa et al., 2021). Given its established role in maintaining oxidative homeostasis, regulating cell proliferation, and mediating apoptosis (Tan et al., 2022), GSTP1 has been further associated with chemoresistance in multiple cancer types, including its involvement in resistance to alkylating agents in breast cancer (Yang et al., 2017) and in promoting metastasis via STAT3 upregulation in colorectal cancer (Wang et al., 2022).

In the context of pancreatic cancer, we demonstrate that GSTP1 is essential for cancer cell survival, as evidenced by the significant reduction in proliferation across PDAC cell lines following its knockdown. These findings are consistent with previous studies showing that GSTP1 is crucial for the proliferation and viability of pancreatic and other cancer cells (Adler, 1999; Louie et al., 2016; Singh et al., 2020; Singh and Reindl, 2021; Wu et al., 2006). Our results further establish that GSTP1 depletion leads to a marked

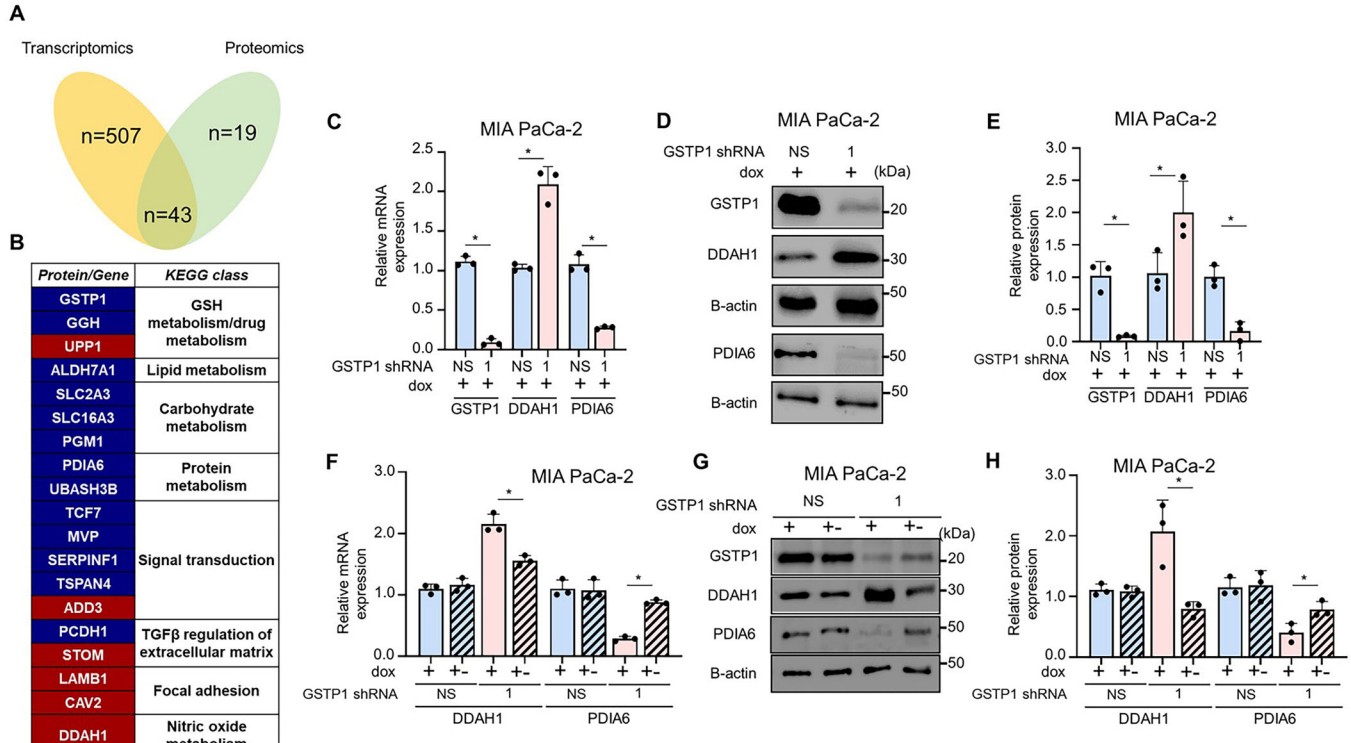

**Fig. 5. RNA sequencing and proteomics experiments reveal parallel responses to GSTP1 knockdown.** (A) Venn diagram showing 43 overlapping differentially expressed genes between the RNA-Seq and proteomics experiments. (B) Summary table of 20 differentially expressed genes with KEGG (Kyoto Encyclopedia of Genes and Genomes) pathway classifications. (C) qRT-PCR validation of PDIA6 and DDAH1 mRNA expression in GSTP1 knockdown MIA PaCa-2 cells. (D,E) Western blot quantification of PDIA6 and DDAH1 protein levels. (F) Reversal of PDIA6 and DDAH1 mRNA expression after 120 h of GSTP1 recovery (dox +-) in MIA PaCa-2 cells. (G,H). Western blotting validation of protein recovery after GSTP1 re-expression. Protein and mRNA levels of PDIA6 and DDAH1 in NS control MIA PaCa-2 cells were compared to those in shGSTP1-1 cells. The results are from three independent experiments (n=3), with significance evaluated by a Student's t-test (*P<0.05). Error bars represent standard deviation.

elevation in ROS levels, which can be reversed upon GSTP1 restoration. This direct link between GSTP1 and ROS regulation suggests that targeting GSTP1 or its associated pathways could serve as a viable therapeutic strategy to selectively induce oxidative stress in PDAC cells, thereby sensitizing them to redox-based therapies.

Metabolic alterations are a hallmark of pancreatic cancer, and emerging evidence highlights dysregulation of metabolic enzymes in PDAC (Wu et al., 2006). The transcriptome and proteome reflect specific metabolic states of PDAC cells as they adapt and shift their metabolic activity and cellular signaling. Insights into these dynamic systems can provide valuable information regarding the molecular changes and mechanisms underlying PDAC progression. Our multiomics analysis provides valuable insights into the molecular consequences of GSTP1 depletion, revealing significant transcriptomic and proteomic shifts in PDAC cells and highlighting pathways related to redox balance, ER stress, and NO metabolism. The identification of 550 differentially expressed genes and 62 differentially expressed proteins underscores the extensive regulatory influence of GSTP1. Notably, the downregulation of 313 genes and 46 proteins suggests that GSTP1 plays a role in maintaining basal expression levels of key cellular regulators, whereas the upregulation of 237 genes and 16 proteins points to potential compensatory mechanisms activated in response to oxidative stress. These findings emphasize the complex interplay between oxidative stress, protein homeostasis, and NO signaling in GSTP1-deficient PDAC cells.

PDIA6 plays a pivotal role in protein folding and ER stress regulation, and its dysregulation has been associated with cancer

progression, including PDAC. Knockdown of PDIA6 has been shown to suppress cell proliferation, invasion, migration, and chemoresistance while enhancing apoptosis in cancer cells (Bai et al., 2019; Cheng et al., 2017; Tao et al., 2023; Yan et al., 2020). Silencing PDIA6 induces ER stress and activates the unfolded protein response (UPR), leading to autophagy and apoptosis (Karamali et al., 2024; Mao et al., 2021). The significant reduction of PDIA6 in GSTP1-knockdown PDAC cells suggests that GSTP1 may regulate PDIA6 expression through transcriptional control, redox-sensitive pathways, or feedback loops involving cellular stress responses, leading to an impaired capacity for protein folding and redox homeostasis. Given PDIA6's role in mitigating ER stress, its downregulation may compromise the ability of PDAC cells to manage oxidative and proteotoxic stress, thereby reducing their survival advantage.

DDAH1 regulated NO metabolism by degrading asymmetric dimethylarginine (ADMA), an endogenous inhibitor of NO synthase. Increased NO levels influence angiogenesis, tumor metabolism, and immune responses (Kami Reddy et al., 2019; Reddy et al., 2018; Ye et al., 2017). Additionally, DDAH1 has been reported to inhibit the epithelial-mesenchymal transition (EMT) through the Wnt/GSK-3β signaling pathway (Boult et al., 2011). The substantial upregulation of DDAH1 in GSTP1-knockdown PDAC cells suggests that cells may compensate for increased oxidative stress by modulating NO production. This response was particularly pronounced at the protein level in HPAF-II cells, pointing to potential post-transcriptional regulatory mechanisms. Importantly, restoring GSTP1 expression reversed both DDAH1 and PDIA6 expression to near-control levels, reinforcing GSTP1's role in regulating redox-sensitive pathways and

**Table 1. Significantly enriched pathways in doxycycline-inducible GSTP1 knockdown MIA PaCa-2 cells as predicted by comparative RNA-Seq analysis**

| Enriched pathway | P-value | q-value |
|---|---|---|
| Metabolism (Homo sapiens) | 8.422E-07 | 2.671E-04 |
| Insulin signaling pathway | 3.702E-06 | 1.794E-04 |
| Extracellular matrix organization | 1.204E-06 | 3.063E-04 |
| Downregulation of TGFβ signaling | 1.646E-05 | 0.002 |
| Downstream signal transduction | 2.054E-05 | 0.002 |
| Signaling by Wnt | 4.922E-05 | 0.003 |
| Prolonged ERK activation/signaling to ERKs | 9.784E-05 | 0.004 |
| Cellular senescence | 1.042E-04 | 0.006 |
| MAPK family cascade signaling | 1.557E-04 | 0.006 |
| Signaling by interleukins | 1.942E-04 | 0.006 |
| Post-translational protein modifications | 2.113E-04 | 0.006 |
| Signaling by NOTCH | 2.414E-04 | 0.006 |
| Interferon alpha/beta signaling | 2.462E-04 | 0.006 |
| $Ca^{2+}$ signaling (Homo sapiens) | 2.537E-04 | 0.006 |
| Glucose metabolism | 4.371E-04 | 0.008 |
| Metabolism of carbohydrates | 5.316E-04 | 0.011 |
| Metabolism of lipids and lipoproteins | 7.241E-04 | 0.013 |
| Purine metabolism | 1.072E-03 | 0.017 |
| Metabolism of nucleotides | 1.308E-03 | 0.019 |
| Metabolism of proteins | 1.384E-03 | 0.021 |
| Insulin receptor signaling cascade | 1.413E-03 | 0.021 |
| Lipid digestion, mobilization, and transport | 3.731E-03 | 0.045 |
| Regulation of insulin secretion | 4.099E-03 | 0.048 |
| Fatty acid, triglycerides, and ketone body metabolism | 4.262E-03 | 0.049 |
| Pyruvate metabolism | 4.352E-03 | 0.049 |

The q-value is an adjusted P-value calculated using the Benjamini-Hochberg method for correcting multiple hypothesis testing.

**Table 2. Significantly enriched pathways in doxycycline-inducible GSTP1 knockdown MIA PaCa-2 cells predicted by the comparative proteomics experiment**

| Enriched pathway | P-value | q-value |
|---|---|---|
| Metabolism | 1.322E-11 | 1.515E-08 |
| Pentose phosphate pathway | 2.561E-08 | 1.464E-05 |
| Tricarboxylic acid (TCA) cycle and respiratory electron transport | 2.603E-07 | 9.891E-05 |
| L1CAM interactions | 6.852E-07 | 1.953E-04 |
| Respiratory electron transport/ATP biosynthesis | 1.838E-06 | 4.191E-04 |
| Immune system | 4.072E-06 | 5.808E-04 |
| Huntington's disease | 4.427E-06 | 5.808E-04 |
| Signaling by NGF | 4.816E-06 | 5.808E-04 |
| Recycling of adhesion molecule L1 | 5.022E-06 | 5.808E-04 |
| Axon guidance | 5.451E-06 | 5.808E-04 |
| Immune system signaling by interferons, interleukins, and growth factors | 9.419E-06 | 6.375E-04 |
| Glucose metabolism | 3.692E-05 | 1.619E-03 |
| Electron transport chain | 5.635E-05 | 2.294E-03 |
| PDGFB signaling | 6.567E-05 | 2.504E-03 |
| Lipid and lipoprotein metabolism | 9.797E-05 | 3.191E-03 |
| Actin cytoskeleton regulation | 1.456E-04 | 4.245E-03 |
| Protein metabolism | 1.773E-04 | 4.594E-03 |
| Mitochondrial pathway of apoptosis | 3.767E-04 | 8.102E-03 |
| Purine metabolism | 4.933E-04 | 9.696E-03 |
| Fas signaling pathway | 5.437E-04 | 0.011 |
| Focal adhesion | 5.632E-04 | 0.011 |
| Activated NOTCH signaling in nucleus | 6.551E-04 | 0.011 |
| Glutathione metabolism | 7.955E-04 | 0.013 |
| MAPK signaling pathway | 9.552E-04 | 0.014 |
| Phospholipid metabolism | 6.461E-03 | 0.048 |

The q-value is an adjusted P-value calculated using the Benjamini-Hochberg method for correcting multiple hypothesis testing.

metabolic adaptation. These findings further suggest that oxidative stress plays a central role in driving transcriptional and translational regulation in GSTP1-deficient PDAC cells.

While this study focused on modeling reversible GSTP1 suppression to better reflect pharmacological inhibition, future investigations will benefit from the development of a complete GSTP1 knockout model using CRISPR/Cas9. Such a model would allow for direct comparisons between partial and total GSTP1 loss, providing a more comprehensive understanding of its essential functions in PDAC. Determining whether complete GSTP1 ablation amplifies effects on cell viability, DDAH1/PDIA6 expression, and oxidative stress would further strengthen the rationale for targeting this pathway therapeutically. Moreover, it would clarify the extent to which residual GST1 expression in knockdown systems may buffer against more severe phenotypes. In addition to the insights uncovered in this study, future investigations will seek to functionally characterize the roles of DDAH1 and PDIA6 in the context of GSTP1 knockdown. Specifically, we are interested in exploring whether knockdown of DDAH1 or overexpression of PDIA6 modulates ROS levels, metabolic adaptations, and cell viability in GSTP1-deficient PDAC cells. Such studies would provide valuable mechanistic understanding of whether these proteins act downstream of GSTP1 and whether their manipulation could enhance the cytotoxic effects observed with GSTP1 depletion.

Redox status influences protein activity through oxidative post-translational modifications, including sulfenylation, nitrosylation, and glutathionylation (Chung et al., 2013; Weerapana et al., 2010). These modifications serve as regulatory switches, modulating protein function in response to fluctuations in oxidative stress. Furthermore, redox status affects cellular oxidative stress, antioxidant capacity, and inflammatory markers, which are critical factors in various pathological conditions (Bouayed et al., 2009). Our data suggest that

increased ROS levels resulting from GSTP1 knockdown downregulate PDIA6, impairing protein folding and redox balance. Simultaneously, DDAH1 upregulation appears to be a redox-sensitive compensatory response aimed at restoring homeostasis via NO metabolism. The intricate interplay among GSTP1, ROS, PDIA6, DDAH1, and NO underscores the complexity of the cellular response to oxidative stress and its implications for cellular proliferation. Overall, the redox status plays a pivotal role in modulating the functions and interactions of proteins involved in redox regulation, cellular signaling, and disease pathogenesis. These insights highlight key vulnerabilities in PDAC's adaptive response to oxidative stress. Further research into the redox-sensitive regulatory mechanisms controlling PDIA6 and DAH1 expression could provide novel therapeutic opportunities.

Our study highlights the critical role of GSTP1 in the survival, metabolic regulation, and redox homeostasis of PDAC cells. GSTP1 depletion leads to significant oxidative stress, triggering compensatory metabolic adaptations, including upregulation of DDAH1 and downregulation of PDIA6. These findings suggest that targeting GSTP1 could be an effective strategy to selectively induce oxidative stress in PDAC cells, potentially enhancing their sensitivity to redox-targeting therapies. Multiomics analysis has proven invaluable in elucidating the molecular effects of GSTP1 depletion, identifying PDIA6 and DDAH1 as key regulators of oxidative stress response in PDAC. By uncovering complex interactions between different omics layers and identifying connections between key molecular players such as PDIA6 and DDAH1, multiomics analysis holds great promise for advancing precision medicine and improving outcomes in pancreatic cancer patients. Our findings open avenues for further research into the mechanisms by which GSTP1 supports PDAC cell survival and proliferation, the signaling pathways that link GSTP1 to NO metabolism and ER stress responses, and the potential of GSTP1

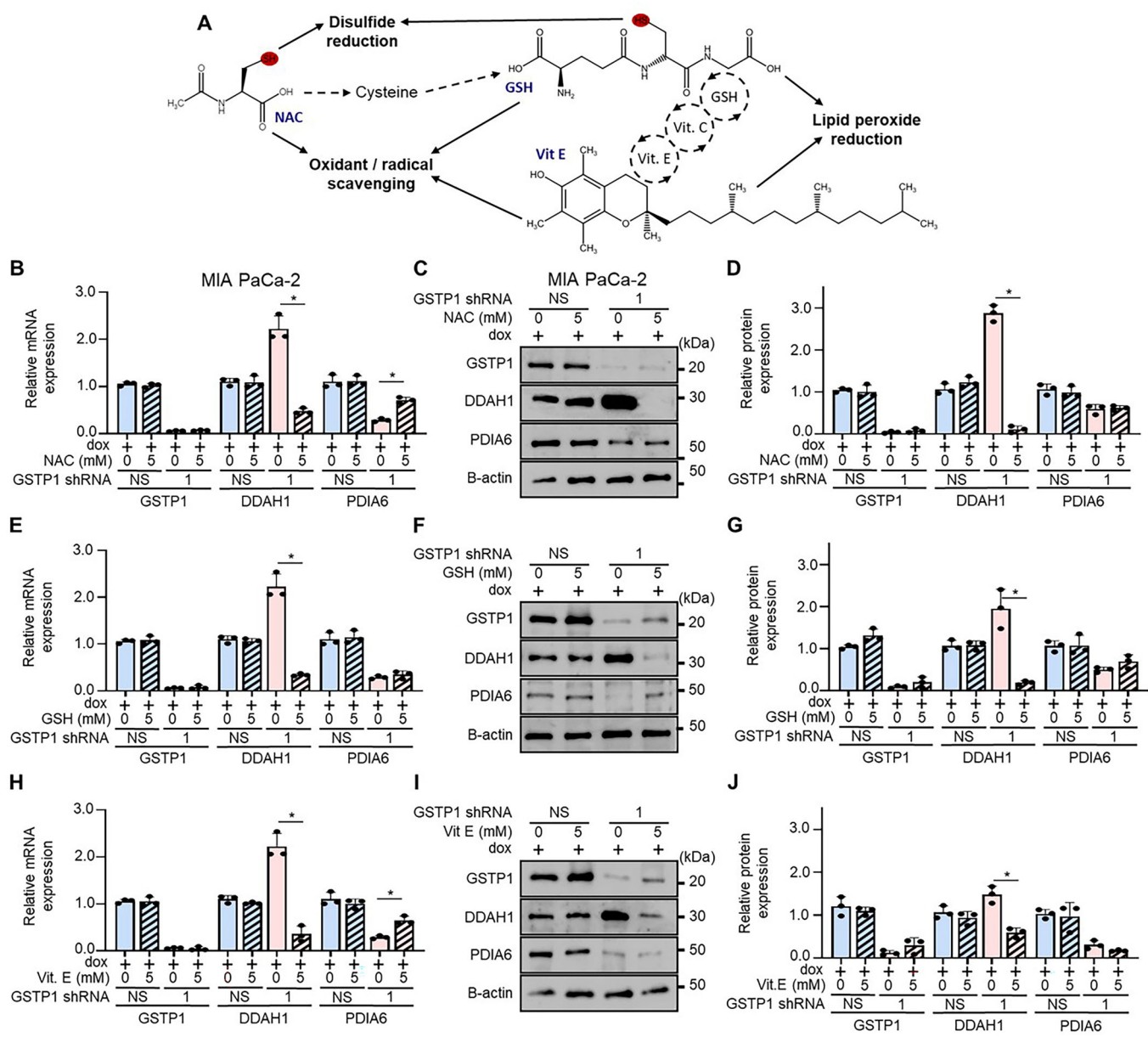

**Fig. 6. Evidence of redox-dependent expression regulation in GSTP1 knockdown cells.** (A) Schematic overview of the antioxidant mechanisms of N-acetyl cysteine (NAC), glutathione (GSH), and Vitamin E. Effects of 48-h treatment with 5 mM NAC on DDAH1 and PDIA6 measured by (B) qPCR and (C-D) western blotting. (E-G) Same as B-D, but with 5 mM GSH treatment. (H-J) Same as B-D, but with 5 mM Vitamin E treatment. PDIA6 and DDAH1 levels in NS control cells were compared to those in GSTP1 knockdown cells (shGSTP1-1). Data are representative of three experiments ($n$=3), with statistical significance assessed by a Student's $t$-test (*$P$<0.05). Error bars represent standard deviation.

inhibition *in vivo*. Additionally, exploring combination therapies that leverage PDAC's oxidative vulnerabilities could enhance therapeutic efficacy. By leveraging multiomics approaches, this study provides a comprehensive perspective on GSTP1's role in PDAC, offering new insights into potential therapeutic interventions. Continued investigation into GSTP1's regulatory networks will be crucial for developing effective treatments for this highly lethal cancer.

## MATERIALS AND METHODS
### Reagents
Puromycin was purchased from Sigma-Aldrich, St. Louis, MO, USA. Doxycycline was purchased from MP Biomedicals, Albany, NY, USA. GSTP1 (sc-376013), PDIA6 (Erp5, sc-365260), and DDAH1 (sc-271337) antibodies were obtained from Santa Cruz Biotechnology, Dallas, TX, USA. GAPDH (97166S) and β-actin (4970S) antibodies, along with

horseradish peroxidase (HRP)-linked anti-mouse (7076S) and anti-rabbit (7074S) IgG secondary antibodies, were supplied by Cell Signaling Technology, Danvers, MA, USA. N-acetylcysteine (NAC) was purchased from Amresco Biochemicals (ELITech Group), Logan, UT, USA. L-Glutathione (GSH) was purchased from Enzo Life Sciences (by VWR), Atlanta, GA, USA. Vitamin E (DL-α-Tocopherol acetate) was received from Sigma-Aldrich. CellROX™ Deep Red Reagent was purchased from Invitrogen (by Thermo Fisher Scientific), Waltham, MA, USA. The MTT reagent was purchased from Thermo Fisher Scientific.

### Cell culture
Human PDAC cell lines (MIA PaCa-2, PANC-1, and HPAF-II) were obtained from the American Type Culture Collection, Manassas, VA, USA. MIA PaCa-2 cells were cultured in Dulbecco's Modified Eagle Medium (DMEM) high-glucose media (GE Healthcare Life Sciences, Chicago, IL, USA) containing 10% (v/v) fetal bovine serum (Atlanta Biologicals,

Flowery Branch, GA, USA) and 2.5% (v/v) horse serum (Corning, Corning, NY, USA). PANC-1 cells were cultured in DMEM high-glucose media containing 10% (v/v) fetal bovine serum (FBS). HPAF-II cells were cultured in Eagle's Minimum Essential Medium (EMEM) (Corning) containing 10% v/v FBS. Cells were maintained at 37°C with 5% CO₂. All cell culture growth media were supplemented with 1% HyClone Antibiotic Antimycotic (Pen/Strep/Fungiezone) Solution (Thomas Scientific, Swedesboro, NJ, USA). At 80% confluency, the cell lines were subcultured using enzymatic digestion with a 0.25% trypsin/1 mM EDTA solution (GE Healthcare Life Sciences). Lentiviral-transfected [non-specific (NS) scramble shRNA control, and GSTP1 knockdown] cells were cultured with 5 μg ml$^{-1}$ puromycin in growth medium to maintain the selection.

## Western blotting

After the cells were washed in cold PBS, a cell culture lysis buffer (Promega, Madison, WI, USA) containing a protease/phosphatase inhibitor cocktail (Cell Signaling Technology) was added. Lysed cells were harvested using cell scrapers and incubated on ice for 30 min. The cell lysate was centrifuged at 13,000 rpm for 10 min at 4°C to collect the protein supernatant. The total protein concentration was measured using the Pierce™ BCA Protein Assay Kit (Thermo Fisher Scientific). The protein samples (10–80 μg) were prepared in Laemmli SDS sample buffer (Thermo Fisher Scientific) with 3-5% BME and subjected to thermal denaturation at 100°C. Samples were loaded in 7-10% SDS-polyacrylamide gels and separated at 100 V for 2.5-3 h at 4°C. Proteins were transferred to a nitrocellulose membrane (GE Healthcare Life Sciences) at 100 V for 70 min at 4°C. Blots were blocked using 5% BSA for 3 h and incubated overnight at 4°C in primary antibody (1:1000). The blots were washed in 1× TBS-T and probed for 1 h at room temperature with a corresponding secondary antibody (1:2000) containing anti-biotin (7075P5) (1:5000). The blots were visualized using SuperSignal West Femto Maximum Sensitivity Substrate (Thermo Fisher Scientific), and chemiluminescence was detected using FluorChem® FC2 Imaging System (Alpha Innotech, San Leandro, CA, USA).

## Quantitative real-time PCR

RNA was isolated from the cells using the Phenol-Free Total RNA Purification Kit (VWR Life Science) according to the manufacturer's protocol. RNA was eluted using 50 μl nuclease-free water. The RNA concentration was measured using the NanoDrop 1000 Spectrophotometer (Thermo Fisher Scientific), and 2 μg of RNA was used to generate cDNA using the qScript cDNA synthesis kit (Quanta Biosciences, Beverly, MA, USA). Real-time qPCR was performed in triplicate using 10 μl PerfeCTa® SYBR® Green Supermix (Quanta Biosciences), 4 μl nuclease-free water, 4 μl of a 1:10 dilution of cDNA, and 1 μl of 3 mM forward and reverse primers (Table S2). Primers were designed using the PrimerQuest™ tool and synthesized by Integrated DNA Technologies, Coralville, IA, USA. The 96-well PCR microplate (Sigma-Aldrich) was run using the Stratagene Mx3000P® Multiplex Quantitative PCR System (Agilent Technologies, Santa Clara, CA, USA) under the following conditions: 95°C for 2 min, followed by 45 cycles of 95°C for 15 s, 55°C for 30 s, and 72°C for 30 s. The results were normalized using β-actin and β-tubulin as housekeeping reference genes, and the data were analyzed using the 2-ΔΔCt method (Livak and Schmittgen, 2001).

## Metabolic cell proliferation assay

MIA PaCa-2 (125/well), PANC-1 (200/well), and HPAF-II (400/well) cells were seeded into 96-well plates. The proliferation of NS control and GSTP1 knockdown PDAC cells (shGSTP1-1 and shGSTP1-2) was measured every 24 h for 10 days by adding 10 μl of 10 mg ml$^{-1}$ 3-(4,5-dimethylthiazol-2-yl)-2,5-diphenyltetrazolium bromide (MTT) reagent to each well and incubating the plates for 3 h at 37°C. The MTT reagent was removed, and DMSO (100 μl/well) was added to solubilize the crystals. The absorbance was measured at 570 nm using a Bio-Rad xMark microplate absorbance spectrophotometer. The data represent the average±s.d. of three independent experiments with 24 technical replicates for each treatment.

To assess the effects of antioxidant treatment on cell proliferation, MIA PaCa-2 NS control and GSTP1 knockdown cells (shGSTP1-1) were seeded and treated with 500 ng ml$^{-1}$ doxycycline for 96 h to induce GSTP1

knockdown. During the final 48 h of doxycycline treatment, cells were treated with 5 mM N-acetylcysteine (NAC), Vitamin E, or Glutathione (GSH). Antioxidants were replenished every 24 h. Before the MTT reagent was added, the media was removed, and the cells were rinsed once with PBS, and then supplied with fresh media to limit potential interactions between the antioxidants and the MTT reagent.

## Detection of ROS

Following treatment with 500 ng ml$^{-1}$ doxycycline for 96 h, NS control and GSTP1 knockdown PDAC cells were harvested and resuspended in complete culture medium. The CellRox™ Deep Red Reagent (Invitrogen) was added to the samples at a final concentration of 1 μM. Unstained controls were included to confirm baseline gating and CellROX specificity. Positive and negative staining controls were prepared using NS control cells treated with 100 μM tert-butyl hydroperoxide (TBHP) for 1 h (positive control) or pretreated with 5 mM NAC for 1 h, followed by 100 μM TBHP for 1 h (negative control). These controls were used to validate assay performance but are not shown in the figures to preserve clarity. The samples were incubated for 60 min at 37°C in an incubator. After staining, the cells were washed once with phosphate-buffered saline (PBS). The samples were immediately analyzed by flow cytometry using a BD Accuri C6 system, using 635 nm excitation. Three technical replicates were included for each experiment, and the experiments were performed three times for each cell line. The data represent the average±s.d. of the fluorescence values from three independent experiments, with three technical replicates for each treatment. FLOWJO software was used to create the histograms.

For experiments assessing the impact of antioxidant treatment on ROS levels, MIA PaCa-2 NS control and GSTP1 knockdown (shGSTP1-1) cells were treated with 500 ng ml$^{-1}$ doxycycline for 96 h, with or without 5 mM NAC co-treatment during the final 48 h. Cells were harvested and resuspended in complete culture medium (with treatments maintained) at a concentration of 2×10⁶ cells per ml. Flow cytometry was performed using a Beckman Coulter CytoFLEX S flow cytometer with 635 nm excitation, and histograms were generated using CytExpert version 2.6. Three independent biological replicates were analyzed per condition.

## RNA-seq analysis

### RNA extraction and sequencing

Following 500 ng ml$^{-1}$ doxycycline treatment for 96 h, total RNA was extracted from four replicates of NS control and shGSTP1-1 MIA PaCa-2 cells using the RNeasy RNA isolation kit (Qiagen, Ann Arbor, MI, USA) according to the manufacturer's instructions. The RNA was quantified using a NanoDrop spectrophotometer (Thermo Fisher Scientific). Four micrograms of total RNA per sample were sent to the University of Minnesota Genomics Center, St. Paul, MN, USA. All samples passed quality control with an RNA integrity number (RIN) of ≥9.4. Unique, dual-indexed, TruSeq stranded mRNA libraries were created. The mean quality score for all libraries was greater than Q30. The libraries were pooled and sequenced in two lanes of the flow cell on an Illumina NovaSeq 6000. The library pool was gel size-selected to have an average insert size of approximately 200 bp.

### Differential gene expression analysis

Reads from each sample were aligned to Ensembl's most updated reference human genome (GRCh38). SAMtools (Li et al., 2009) was used to generate, sort, and index BAM (binary alignment/map) files. Gene expression was calculated as the total number of reads for each sample that were uniquely aligned to the reference genome and binned by gene coordinate annotations. The generated reads were assigned genomic features by using the featureCounts function. Differential gene expression analysis was performed between NS control and shGSTP1-1 (GSTP1 knockdown) MIA PaCa-2 cells using the Bioconductor package DESeq2 (Love et al., 2014). To account for differences in sequencing depth across samples, raw read counts were normalized using the methodologies implemented in DESeq2. Differential expression of the normalized read counts was assessed using the negative binomial test with the Benjamini-Hochberg false discovery rate (FDR) adjustment method, as implemented by DESeq2. For our analysis, an FDR of 0.05 was applied, and genes with a p-adjusted

value of less than or equal to 0.05 and a log$_2$-fold change of less than −1 or greater than +1 were defined as significantly down- or upregulated. Pathway analyses were performed using Ingenuity Pathway Analysis (IPA), Enrichr (Kuleshov et al., 2016), and Reactome (Jassal et al., 2019) tools.

## Proteomics
### Sample preparation
NS control and shGSTP1-1 MIA PaCa-2 cells were treated with 500 ng ml$^{-1}$ doxycycline for 96 h. Five million cells from NS control and shGSTP1-1 MIA PaCa-2 cells were collected for a mass spectrometry (MS)-based proteomics experiment. The samples were stored at −80°C until the proteomics experiment. Five biological replicates from each group were sent to the Proteomics Core Facility at the University of Arkansas for Medical Sciences, Little Rock, AR, USA.

### Tandem mass tag (TMT) labeling and HPLC analysis
Total protein was extracted at the Proteomics Core Facility of the University of Arkansas for Medical Sciences. Proteins were reduced, alkylated, and purified by chloroform/methanol extraction prior to digestion with sequencing-grade modified porcine trypsin (Promega). Tryptic peptides were labeled using tandem mass tag isobaric labeling reagents (Thermo Fisher Scientific) according to the manufacturer's instructions and combined into one 10-plex sample group. The labeled peptide multiplex was separated into 46 fractions on a 100×1.0 mm Acquity BEH C18 column (Waters, Milford, MA, USA) using an UltiMate 3000 UHPLC system (Thermo Fisher Scientific) with a 50 min gradient from 99:1 to 60:40 buffer A:B ratio under basic pH conditions, and then consolidated into 18 super-fractions. Each super-fraction was further separated by reverse phase XSelect CSH C18 2.5 µm resin (Waters) on an in-line 150×0.075 mm column using an UltiMate 3000 RSLCnano system (Thermo Fisher Scientific). Peptides were eluted using a 60 min gradient from a 98:2 to 60:40 buffer A:B ratio. Eluted peptides were ionized by electrospray (2.2 kV), followed by mass spectrometric analysis on an Orbitrap Eclipse Tribrid mass spectrometer (Thermo Fisher Scientific) using multi-notch MS3 parameters. MS data were acquired using the FTMS analyzer in top-speed profile mode at a resolution of 120,000 over a range of 375 to 1500 m z$^{-1}$. Following CID activation with a normalized collision energy of 35.0, MS/MS data were acquired using the ion trap analyzer in centroid mode and normal mass range. Using synchronous precursor selection, up to 10 MS/MS precursors were selected for HCD activation with a normalized collision energy of 65.0, followed by the acquisition of MS3 reporter ion data using the FTMS analyzer in profile mode at a resolution of 50,000 over a range of 100-500 m z$^{-1}$. Buffer A=0.1% formic acid and 0.5% acetonitrile. Buffer B=0.1% formic acid and 99.9% acetonitrile. Both buffers were adjusted to pH 10 with ammonium hydroxide for offline separation.

### Differential protein expression analysis
Protein TMT MS3 reporter-ion intensity values were assessed for quality using the in-house ProteiNorm app (University of Arkansas for Medical Sciences), a user-friendly tool for systematic evaluation of normalization methods, imputation of missing values, and comparison of different differential abundance methods. Popular normalization methods were evaluated, including log$_2$ normalization, median normalization, mean normalization, variance-stabilizing normalization (Huber et al., 2002), quantile normalization (Bolstad, 2021), cyclic loess normalization (Ritchie et al., 2015), global robust linear regression normalization (RLR) (Chawade et al., 2014), and global intensity normalization (Chawade et al., 2014). The individual performance of each method was evaluated by comparing the following matrices: total intensity, pooled intragroup coefficient of variation (PCV), pooled intragroup median absolute deviation (PMAD), pooled intragroup estimate of variance (PEV), intragroup correlation, sample correlation heatmap (Pearson), and log$_2$-ratio distributions. The normalized data were used to perform statistical analysis using Linear Models for Microarray Data (limma) with empirical Bayes (eBayes) smoothing to the standard errors (Ritchie et al., 2015). Proteins with an FDR-adjusted P-value <0.05 and a fold change >2 were considered significant. Pathway analyses were performed using Ingenuity Pathway Analysis (IPA), Reactome (Jassal et al., 2019), and Enrichr (Kuleshov et al., 2016) tools.

## Statistical analysis
The results of at least three biological replicate experiments, each with at least three technical replicates, are presented as the mean±s.d. The significance between the two groups (NS control and shGSTP1 knockdown) was determined using a Student's t-test, with a P-value <0.05 considered statistically significant, except for the interpretation of proteomics and transcriptomics experiments (FDR-adjusted P-value <0.05). Cell proliferation was analyzed separately for each PDAC cell line with the knockdown line, time, and experimental replicates as factors. Statistical analyses were performed using GraphPad Prism 10, version 10.4.0.

## Acknowledgements
The authors would like to thank the University of Minnesota Genomics Center, St. Paul, MN, and the Proteomics Core Facility at the University of Arkansas for Medical Sciences, Little Rock, AR, for providing resources that contributed to the research results reported in this paper. We also acknowledge Jeffrey Kittilson and Dr Philip Salu for their technical support and assistance. Special thanks to Scott Hoselton at the NDSU Dr Thomas Glass Biotech Innovation Core for his support with flow cytometry experiments.

## Competing interests
The authors declare no competing or financial interests.

## Author contributions
Conceptualization: J.D., R.R.S., K.M.R.; Data curation: J.D., R.R.S., K.S.; Formal analysis: J.D.; Funding acquisition: K.M.R.; Investigation: J.D.; Methodology: J.D., R.R.S., K.S.; Project administration: K.M.R.; Supervision: K.M.R.; Validation: J.D., K.S.; Visualization: J.D.; Writing – original draft: J.D.; Writing – review & editing: J.D., R.R.S., K.S., K.M.R.

## Funding
This research was supported by the National Institutes of Health (NIH) grants 1R15CA249714 and 1P20GM109024 awarded to K.M.R. The contents of this study are solely the responsibility of the authors and do not necessarily represent the official views of the NIH. Open Access funding provided by North Dakota State University. Deposited in PMC for immediate release.

## Data and resource availability
All data supporting the findings of this study are included within this manuscript and its supplementary material. Additional datasets or specific raw data supporting the conclusions of this study are available from the corresponding author upon reasonable request.

## Peer review history
The peer review history is available online at https://journals.biologists.com/bio/lookup/doi/10.1242/bio.061986.reviewer-comments.pdf.

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
