## [Peer Review File · Biology Open]

Multimomics Analysis of GSTP1 Knockdown Pancreatic Cancer Cells Reveals Key Regulators of Redox and Metabolic Homeostasis

Rahul Raj Singh, Katherine Schmidt, Katie M. Reindl and Duttenhefner Jenna
DOI: 10.1242/bio.061986

Editor: Chris Maher

Review timeline

Original submission:	18 March 2025
Editorial decision:	22 April 2025
First revision received:	11 July 2025
Accepted:	17 July 2025

Original submission

First decision letter

MS ID#: bio.061986

MS Title: Multimomics Analysis of GSTP1 Knockdown Pancreatic Cancer Cells Reveals Key Regulators of Redox and Metabolic Homeostasis

Authors: Jenna Duttenhefner; Rahul Raj Singh; Katherine Schmidt; Katie M. Reindl

Dear Dr Duttenhefner,

I have now reached a decision on the above manuscript.

The reviewer reports are shown at the bottom of this email or can be accessed, together with a copy of this decision letter, by going to:

As you will see, the reviewers gave favourable reports, but raised some critical points that will require amendments to your manuscript. I hope that you will be able to carry these out, because we would like to be able to accept your paper.

Reviewer 1

Comments for the author

The authors present a well executed study investigating the cellular effects of GSTP1 knockdown in pancreatic cancer cell lines. A few revisions and additional controls would strengthen the study.

-The authors use an inducible knockdown system of GSTP1 which has certain advantages, however there is residual protein expression that may be masking phenotype of full loss and explain the rather modest effect on cell viability. Have the authors considered making a CRISPR knockout? Are cells able to survive without any residual GSTP1? A full knockout would ideally be added to this study and at least used for a few experiments (viability, DDAH1/PDIA6 levels).

-When do the changes in ROS appear? Did the authors do a timecourse? It would be nice to see how this compares to the timing used for transcriptomic/proteomic analysis.

- The authors use antioxidant small molecules in figure 6 to rescue DDAH1/PDIA6 levels. It would be nice to see how these small molecules effect the viability go GSTP1 KD cells (figure 1) and the ROS signature (figure 2) - does antioxidant use rescue the phenotype?
- A positive control that generates ROS (such as H2O2) should be used in the CellROX assay.
- The authors focus on the connection of GSTP1 knockdown to DDAH1 and PDIA6 levels. It would be nice to include knockdown of DDAH1/overexpression of PDIA6 in the setting of GSTP1 KD to see how this affects cell viability and ROS. For example, does combination knockdown of GSTP1 and DDAH1 lead to enhanced cell death and increased ROS/altered metabolism?

Reviewer 2

Comments for the author

A convincing study investigating inducible GSTP1 repression at the cellular level. The systematics, methodology, and graphical presentation are entirely satisfactory. I could only identify a few spelling errors. In sum, I recommend accepting the thesis.

Minor issues:

Line 345: (10-80µg) - not space - (10 - 80 µg)

Line 393: 500ng/mL - 500 ng/ml

Reviewer's Responses to Questions

Experimental quality

Does each figure have the proper controls?

If 'No', please indicate reasons in Comments for Author box below.

Reviewer #1:

- Yes

Reviewer #2:

- No

Were the data analyzed using appropriate statistical tests?

If 'No', please indicate reasons in Comments for Author box below.

Reviewer #1:

- Yes

Reviewer #2:

- Yes

Reproducibility

Were experiments performed using adequate number of biological replicates?

If 'No', please indicate reasons in Comments for Author box below.

Reviewer #1:

- Yes

Reviewer #2:

- No

Does the methods section provide sufficient detail to permit reproducibility?
If 'No', please indicate reasons in Comments for Author box below.

Reviewer #1:

- Yes

Reviewer #2:

- Yes

Completeness

Are the manuscript's conclusions supported by the data?
If 'No', please indicate reasons in Comments for Author box below.

Reviewer #1:

- Yes

Reviewer #2:

- No

Scholarship

Do the authors cite and discuss the merits of data that would argue for and against their conclusion?
If 'No', please indicate reasons in Comments for Author box below.

Reviewer #1:

- Yes

Reviewer #2:

- No

Does the manuscript title & abstract accurately reflect the contents of the manuscript, without hyperbole?
If 'No', please indicate reasons in Comments for Author box below.

Reviewer #1:

- Yes

Reviewer #2:

- No

First revision

Author response to reviewers' comments

Reviewer 1:

Reviewer 1: The authors present a well executed study investigating the cellular effects of GSTP1 knockdown in pancreatic cancer cell lines. A few revisions and additional controls would strengthen the study.

1. The authors use an inducible knockdown system of GSTP1 which has certain advantages, however there is residual protein expression that may be masking phenotype of full loss and explain the rather modest effect on cell viability. Have the authors considered making a CRISPR knockout? Are cells able to survive without any residual GSTP1? A full knockout would ideally be added to this study and at least used for a few experiments (viability, DDAH1/PDIA6 levels).

JD - We have considered using CRISPR/Cas9 to generate a complete GSTP1 knockout model; however, our decision to use an inducible knockdown system was intentional and aligns with our study's primary goal—to model the effects of partial and reversible GSTP1 suppression, which more closely mimics pharmacological inhibition. While CRISPR knockouts can offer insight into the complete loss of a gene, our inducible system allows us to temporally control GSTP1 expression and investigate the dynamic cellular responses to its depletion and restoration, minimizing potential compensatory effects that may confound interpretation in permanent knockout models.

Moreover, GSTP1 knockout mice and double knockout models in the literature have shown viability, likely due to redundancy within the GST family and other detoxification enzymes. These findings support the notion that complete loss of GSTP1 is survivable in certain contexts, although we acknowledge that the absence of total GSTP1 in our study may result in a more modest phenotype. Nevertheless, our model effectively captures key redox, metabolic, and transcriptomic changes relevant to therapeutic targeting, and we believe it remains a strong and clinically relevant tool for understanding the mechanistic roles of GSTP1 in PDAC. We agree that generating a GSTP1 knockout model would be valuable for future studies and have addressed this possibility explicitly in the revised Discussion section (lines: 302-310)

2. When do the changes in ROS appear? Did the authors do a timecourse? It would be nice to see how this compares to the timing used for transcriptomic/proteomic analysis.

JD - While a time course was initially conducted to explore the dynamics of ROS accumulation following GSTP1 knockdown, all ROS measurements presented in this manuscript were performed at the 96-hour timepoint. This timepoint was selected to align with the transcriptomic and proteomic analyses, ensuring consistency across experimental platforms (Methods, line 426). We chose to present only the 96-hour data for clarity and direct comparability with the multiomics datasets.

3. The authors use antioxidant small molecules in figure 6 to rescue DDAH1/PDIA6 levels. It would be nice to see how these small molecules effect the viability go GSTP1 KD cells (figure 1) and the ROS signature (figure 2) - does antioxidant use rescue the phenotype?

JD - We appreciate the reviewer's suggestion and have performed additional experiments to

assess the impact of antioxidant treatment on both viability and ROS levels in GSTP1 knockdown cells. Specifically we conducted MTT assays and ROS flow cytometry following treatment with N-acetylcysteine (NAC), glutathione (GSH), and Vitamin E. These new data are now included in Supplementary Figure 5 (Fig. S5) and described in the Methods (MTT, line 418; flow cytometry, line 441) and Results (lines 214-238). While antioxidant treatment did not significantly restore viability in GSTP1-deficient cells, NAC treatment effectively reduced ROS levels to baseline, validating the activity of the antioxidants and supporting our interpretation of the oxidative phenotype.

4. A positive control that generates ROS (such as H₂O₂) should be used in the CellROX assay.

JD - We thank the reviewer for this important point. As part of our ROS flow cytometry assay, we included a positive control using 100 μ M tert-butyl hydroperoxide (TBHP) to induce ROS in NS control cells, as well as a negative control in which cells were pretreated with 5 mM NAC for 1 hour prior to TBHP exposure. These controls were used in each experiment to confirm CellROX Deep Red sensitivity and validate assay performance. While these controls are not shown in the manuscript to maintain figure clarity, they are described in the Methods section (line 430).

5. The authors focus on the connection of GSTP1 knockdown to DDAH1 and PDIA6 levels. It would be nice to include knockdown of DDAH1/overexpression of PDIA6 in the setting of GSTP1 KD to see how this affects cell viability and ROS. For example, does combination knockdown of GSTP1 and DDAH1 lead to enhanced cell death and increased ROS/altered metabolism?

JD - We thank the reviewer for this insightful suggestion. This is an important next step that we have considered as a potential future direction for our work. While the current study was designed to identify molecular changes associated with GSTP1 knockdown, we agree that functional experiments involving modulation of DDAH1 and PDIA6, such as DDAH1 knockdown and PDIA6 overexpression in the context of GSTP1 loss, would provide valuable mechanistic insight into how these proteins contribute to redox balance and cell viability. In response to the reviewer's comment, we have expanded our discussion to explicitly highlight this as a key future direction [Lines 310 - 316]. We believe such experiments would be instrumental in determining whether these genes act downstream of GSTP1 and if their manipulation could modulate ROS levels or synergize to enhance cytotoxic effects in PDAC cells.

Reviewer 2:

A convincing study investigating inducible GSTP1 repression at the cellular level. The systematics, methodology, and graphical presentation are entirely satisfactory. I could only identify a few spelling errors. In sum, I recommend accepting the thesis.

Minor issues:

1. Line 345: (10-80 μ g) - not space - (10 - 80 μ g)

JD - Thank you for noting this formatting issue. The spacing has been corrected to read "(10-80 μ g)" now at line 386.

2. Line 393: 500ng/mL - 500 ng/ml

JD - The spacing has been corrected to "500 ng/mL" now at line 423.

Second decision letter

MS ID#: bio.061986R1

MS Title: Multiomics Analysis of GSTP1 Knockdown Pancreatic Cancer Cells Reveals Key Regulators of Redox and Metabolic Homeostasis

Authors: Jenna Duttenhefner; Rahul Raj Singh; Katherine Schmidt; Katie M. Reindl

Dear Dr Duttenhefner,

I am happy to tell you that your manuscript has been accepted for publication in Biology Open, pending our standard publication integrity checks. It was accepted on 17 July 2025.